# Glucose Metabolism and Aging of Hematopoietic Stem and Progenitor Cells

**DOI:** 10.3390/ijms23063028

**Published:** 2022-03-11

**Authors:** Laura Poisa-Beiro, Jonathan J. M. Landry, Simon Raffel, Motomu Tanaka, Judith Zaugg, Anne-Claude Gavin, Anthony D. Ho

**Affiliations:** 1Department of Medicine V, University of Heidelberg, Im Neuenheimer Feld 410, 69120 Heidelberg, Germany; laura.poisabeiro@med.uni-heidelberg.de (L.P.-B.); simon.raffel@med.uni-heidelberg.de (S.R.); 2Molecular Medicine Partnership Unit Heidelberg, European Molecular Biology Laboratory (EMBL) & Heidelberg University, 69120 Heidelberg, Germany; judith.zaugg@embl.de (J.Z.); anne-claude.gavin@unige.ch (A.-C.G.); 3Genomics Core Facility, European Molecular Biology Laboratory (EMBL), Meyerhofstr. 1, 69117 Heidelberg, Germany; jonathan.landry@embl.de; 4Physical Chemistry of Biosystems, Inst, Heidelberg University, Im Neuenheimer Feld 253, 69120 Heidelberg, Germany; tanaka@uni-heidelberg.de; 5European Molecular Biology Laboratory, Meyerhofstr. 1, 69117 Heidelberg, Germany; 6Department for Cell Physiology and Metabolism, Centre Medical Universitaire, University of Geneva, Rue Michel-Servet 1, 1211 Geneva, Switzerland

**Keywords:** hematopoietic stem and progenitor cells, aging, senescence signature, central carbon metabolism, glycolysis

## Abstract

Comprehensive proteomics studies of human hematopoietic stem and progenitor cells (HSPC) have revealed that aging of the HSPC compartment is characterized by elevated glycolysis. This is in addition to deregulations found in murine transcriptomics studies, such as an increased differentiation bias towards the myeloid lineage, alterations in DNA repair, and a decrease in lymphoid development. The increase in glycolytic enzyme activity is caused by the expansion of a more glycolytic HSPC subset. We therefore developed a method to isolate HSPC into three distinct categories according to their glucose uptake (GU) levels, namely the GU^high^, GU^inter^ and GU^low^ subsets. Single-cell transcriptomics studies showed that the GU^high^ subset is highly enriched for HSPC with a differentiation bias towards myeloid lineages. Gene set enrichment analysis (GSEA) demonstrated that the gene sets for cell cycle arrest, senescence-associated secretory phenotype, and the anti-apoptosis and P53 pathways are significantly upregulated in the GU^high^ population. With this series of studies, we have produced a comprehensive proteomics and single-cell transcriptomics atlas of molecular changes in human HSPC upon aging. Although many of the molecular deregulations are similar to those found in mice, there are significant differences. The most unique finding is the association of elevated central carbon metabolism with senescence. Due to the lack of specific markers, the isolation and collection of senescent cells have yet to be developed, especially for human HSPC. The GU^high^ subset from the human HSPC compartment possesses all the transcriptome characteristics of senescence. This property may be exploited to accurately enrich, visualize, and trace senescence development in human bone marrow.

## 1. Introduction

The regenerative power of a living organism is reflected by the potential of its somatic stem cells to replace damaged tissues, especially in a biological system that is characterized by a high cell turnover such as the hematopoietic system [1,2]. A living organism is hence as old as its somatic stem cells [3].

In murine models, aging of the hematopoietic system is reflected by a decrease in regenerative potential and in the competence of the adaptive immune system [4,5,6,7,8,9,10]. The hematopoietic stem and progenitor cells (HSPC) undergo quantitative and functional changes with age, resulting in a diminished engraftment potential [4,5,6,7,8,9]. Decline of the immune system is associated with the propensity to develop cancers. This degeneration is a sum product of interaction between HSPC and the cellular determinants in the bone marrow niche. The latter has been shown to significantly contribute to the decline in HSPC function over time and include mesenchymal stromal cells (MSC), endothelial cells of the vascular system, skeletal stem cells, as well as macrophages/monocytes and lymphocytes [10,11,12,13,14,15,16,17]. However, transplantation of aged HSPC into young mice was not able to improve regenerative function [4,7,18], suggesting the significant role of intrinsic factors in the aging HSPC.

Studies on HSPC in murine aging models have advanced our knowledge of the mechanisms of the aging process. Present evidence indicates that aging involves a gradual, heterogeneous deficiency in the structure and function of a subpopulation of HSPC that have become senescent [19]. With the advances made in multi-omics analysis, HSPC from young mice have been compared with aged mice and the differential gene expression profiles of aged HSPC characterized [20]. Transcriptomics studies have provided a blueprint of the underlying molecular mechanisms and indicated that genes associated with DNA damage and repair, myeloid lineage specification, as well as with myeloid malignancies were upregulated in old HSPC [7,19].

While most of the current knowledge on mechanisms of aging has been gained from animal models [21], similar studies on human HSPC have been scarce [22,23]. In mice, elimination of aged HSPC has been shown to induce a revival of normal HSPC proliferation and differentiation, leading to a rejuvenation of the hematopoietic compartment [24,25]. It is, however, not clear if the knowledge gained in murine models of aging can be extrapolated to human systems.

In this review, we will focus on the role of intrinsic factors that contribute to HSPC aging, under consideration of the recent knowledge gained from comprehensive proteomics and single-cell transcriptomics studies. Following the cue provided by multi-omics studies, the center of attention will be on the metabolic deregulation associated with aging HSPC and their potential relevance for “senolysis” therapy strategies.

## 2. Aging and Senescence

Whereas aging is defined as the progressive changes in an organism that lead to a decline of biological functions needed for survival, senescence is defined as an irreversible form of long-term cell cycle arrest induced by intracellular or extracellular damage in individual cells [19,26,27,28,29,30]. Senescence is a consequence of injuries such as genomic instability and telomere attrition [19,31,32]. The purpose of senescence is to limit the proliferation of damaged cells and to protect the organism from malignant cell transformation. Senescence also plays an essential and physiological role during normal development such as embryogenesis and is needed for tissue homeostasis [33]. However, accumulation of senescent cells in the stem cell compartment leads to detrimental effects, with an overall decline in regenerative potential [19,29,34,35].

Multi-omics studies of HSPC in murine models for aging have indicated that a small number of senescent cells within the HSPC compartment are responsible for all the consequences of aging. Within the physiological aging process, factors that initiate cellular senescence include telomere attrition, oxidative stress, DNA damage and metabolic dysfunction [19,29,36].

## 3. Lessons Learned from Murine Models of Aging

Studies on aging of the hematopoietic system in mice have demonstrated that the number of primitive HSPC, defined as Thy-1^lo^Sca-1^hi^Lin Mac-1^−^CD4^−^c-kit^+^cells [4] or “cobblestone area forming cells” [5] in the bone marrow increased by two- to ten-fold in aged animals [4,5,7,8,9]. However, these HSPC exhibited functional defects such as diminished regenerative potential in serial transplantation assays. Transplantation of aged HSPC into young mice was not able to improve regenerative function [4,7], suggesting the significance of intrinsic factors in the aging HSPC. This decline in engraftment potential with age is supported by clinical studies demonstrating that HSPC harvested from elderly donors are associated with a decreased transplantation success rate [37]. The increase in phenotypically primitive HSPC may be a compensatory mechanism to overcome their loss in function.

Transcriptomics studies comparing aged and young murine HSPC have indicated lineage skewing of HSPC towards myeloid differentiation at the cost of lymphoid development [21,38], delays in cell cycling and irreversible growth arrest [39,40,41], elevated inflammation and stress responses [42,43], replicative stress [44], and chromatin remodelling and DNA hypomethylation [42,45,46,47,48] in aged HSPC. Correlating transcriptomics with cellular function in murine HSPC at a single-cell level, Grover et al. showed that an increase in platelet priming was the predominant age-dependent change [38] and this might contribute to the age-associated decrease in lymphopoiesis.

To determine whether a consistent aging signature could be derived from published and unpublished transcriptomics studies, Svendsen et al. have reviewed 16 datasets comparing the transcriptomes of old versus young murine HSPC [20]. They detected an overall increase in transcriptional activation in aged HSPC, and established that the majority of the deregulated genes upon aging are membrane associated. Some of the deregulated genes are directly involved in the functional decline. With this meta-analysis, they have provided a reference atlas of the transcriptomics landscape for aging of murine HSPC. This group of authors have also demonstrated that the aged HSPC pool is heterogeneous, with the presence of “young-like” HSPC in aged bone marrow [20].

Applying immunophenotypic analysis and gene expression profiling of human primitive HSPC from different age groups, Pang et al. demonstrated that HSPC increased in frequency with age, were less quiescent, and exhibited myeloid-biased differentiation potential [22]. Moreover, genes associated with cell cycle, myeloid lineage specification and myeloid malignancies were upregulated in elderly human HSPC, confirming some of the observations found in mice.

## 4. Comprehensive Proteomics Approach—Major Findings

Using an unbiased, comprehensive proteomic approach, our group has analyzed the molecular features of aging of HSPC as well as five other cell types that constitute the niche in human bone marrow. We have presented an atlas of the age-associated alterations in the proteomics landscapes of human HSPC, as well as subpopulations comprising the bone marrow niche derived from 59 human subjects (age range 20–60 years), i.e., across the time span of 40 years [23]. The results of proteomics studies are summarized in Figure 1 (modified from [23]).

Among the significant changes in abundance of proteins with age in the HSPC, we found a significant age-associated increase in enzymes catalyzing the upper part of glycolysis: hexokinase 1 (HK1), phosphofructokinase M (PFKM), as well as the glycolytic enzymes aldolase C (ALDOC) and triosephosphate isomerase 1 (TPI1). The principal findings are demonstrated in Figure 2 (from [23]). The enzymes involved in glycogen metabolism, glycogen phosphorylases brain and liver form (PYGB, PYGL), and glycogen debranching enzyme (AGL) were significantly more abundant in old HSPC. Compatible with these changes, an increase in abundance of trans-aldolase 1 (TALDO1), glycerol-3-phosphate dehydrogenase (GPD2), and dihydroxy-acetone kinase (DAK) provides further evidence for elevated activity in the upper part of the central carbon metabolism in aging HSPC (see Figure 2). The second phase of central carbon metabolism, characterized by the production of ATP, NADH and pyruvate and proteins involved in the Krebs cycle, remained unaffected. This upregulation of the upper part of the glycolytic pathway is unique for the HSPC compartment and is not found in the other five cell types of the niche, including MSC, as summarized in Figure 2.

Other significant changes found with the proteomics study include an increase in synthesis of prostaglandins and thromboxanes, in arachidonic acid metabolism, in metabolism of nitric oxide and in proteins that are associated with myeloid development in older HSPC (Figure 1). There was a remarkable decrease in abundance for proteins responsible for lymphoid development, for stabilization of DNA replication and for methylation processes. The enhanced metabolic and anabolic activity of aged HSPC, especially of central carbon metabolism is, however, the most novel and unique finding using a proteomics approach.

## 5. Elevated Central Carbon Metabolism in HSPC upon Aging

To validate our proteomics results, we have also assessed and compared the enzymatic activities of representative glycolytic enzymes hexokinase 1 (HK 1), phosphofructokinase M, aldolase C (ALDO C) and triosephosphate isomerase 1 (TPI 1) in the CD34+ cells derived from old versus young human subjects. In addition, activity of adenylate kinase (ADK), an enzyme regulating glucose-driven energy metabolism was also examined. Our data demonstrated that the activities of ADK, TPI and HK were significantly increased in the elderly cohort [49], validating the results from the proteomics studies.

Glucose metabolism has been shown to play a pivotal role in governing stem cell fate in terms of proliferation, differentiation, or dormancy on the one hand [50], and to influence chromatin structure and transcription on the other [51]. Our proteomics and functional studies have provided evidence for a prominent shift in metabolism of human HSPC during the aging process [23,49]. The upregulation of the preparatory phase of glycolysis, of fatty acid oxidation, and of acetyl-CoA metabolism, with redirection of glycolytic carbons to pathways that branch out of the Krebs cycle for anabolic processes, i.e., for nucleotides, lipids, amino acids synthesis, was reminiscent of the Warburg effect [51,52,53].

The question then arises whether this increase in glycolytic enzyme levels is caused by an increase in these enzymes on a per-cell basis upon aging, or by the expansion of one of the HSPC subsets that have become more glycolytic than the others.

We have assessed the changes in glycogen content in HSPC from young (19–31 years in this study) and old (59–71 years in this study) human subjects via a semi-quantitative PAS analysis [49]. The results are summarized in Figure 3 and Table 1. For older human subjects, the average glycogen content was 3.5-fold higher and was highly significant (one-sided t-test, *p* = 1.2 × 10^−6^) (Figure 3a). For young human subjects, the glycogen content was relatively homogeneous and most CD34+ cells showed intermediate and low glucose uptake. There were scarcely any CD34+ cells with the GU^high^ phenotype in young subjects.

Table 1 shows a list of the fractions of GU^high^, GU^inter^, and GU^low^ CD34+ cells from young (<35 years, *n* = 3) versus older (>50 years; *n* = 7) human subjects. Whereas there is no difference in the percentages of GU^inter^ and GU^low^ fractions between young and older subjects, the difference in GU^high^ is significant (*p* = 0.02, one-sided *t*-test).

Single-cell RNA sequencing (scRNA-seq) analysis of aged CD34+ cells in comparison to young CD34+ has provided further evidence that the re-wiring of central carbon metabolism was found only in the myeloid-primed subpopulation of HSPC and not in the lymphoid-primed cells, nor in the CD34+ subset that showed no lineage commitment [23,49].

Our results have therefore provided evidence that the upregulation of glycolytic enzymes is caused by the relative expansion of a subset of CD34+ cells that have become more glycolytic [49].

## 6. Separation of HSPC According to Carbon Metabolic Levels

Following the queue that a small population of HSPC with elevated central carbon metabolism might be responsible for the aging HSPC phenotype in humans, we developed a method to isolate subsets of HSPC according to their levels of carbon metabolism, using glucose uptake (GU) as a surrogate marker [49].

Given the knowledge gained from the semi-quantitative PAS reactivity and functional enzymatic assays, the variations in glycolytic metabolism could be exploited to isolate the HSPC according to their respective GU capacity. From the bone marrow of older human subjects, we were consistently able to separate the HSPC into GU^high^, GU^inter^ and GU^low^ subpopulations. The same is true for adults younger than 35 years, with the exception that the GU^high^ subset was hardly detectable. After separation of CD34+ cells according to glucose metabolic levels, the GU^high^, GU^inter^ and GU^low^ subsets were harvested for further characterization using single-cell RNA sequencing (scRNA-seq) technology.

## 7. Single-Cell Transcriptome Studies of HSPC with Different Levels of Carbon Metabolism

We have performed scRNA-Seq studies on the HSPC (CD34+ cells) derived from 10 healthy human subjects (old adults *n* = 6; age range: 62–74 years, young adults *n* = 4; age range: 21–33 years) with special focus on the comparisons between GU^high^ versus GU^low^ and GU^inter^ from each individual [54]. For the comparisons of HSPC with different levels of glucose metabolism, 196 cells from each of the categories GU^high^, GU^inter^ and GU^low^ isolated from each individual human subject tested were sequenced and the transcriptome profiles compared. As a reference, 196 CD34+ cells from individual old subjects were compared with 196 CD34+ cells derived from individual young human subjects [49,54].

### 7.1. DNA Damage, Telomere Attrition, and Cell Cycle Arrest

Damage of the telomeres represents one of the major injuries that induce senescence upon physiological aging [31]. The telomere region is protected by a multi-protein complex shelterin, with POT1 and TERF2 as major components [19,55]. TERF2 and POT1 act independently to repress either the ATM or ATR kinase signaling pathway required for efficient non-homologous end-joining of dysfunctional telomeres [55,56]. A deficiency in TERF2 or POT1 induces genome instability by shortening telomeres and causes DNA damage, which in turn activates a kinase cascade involving first ATM and ATR and then CHEK1 and CHEK2, resulting in sustained p53 activation [57,58]. The latter then leads to a permanent arrest of the cell cycle by persistent induction of the cyclin-dependent kinase inhibitor CDKN1A (P21CIP1), resulting in hypo-phosphorylated RB and ultimately in cell cycle exit [58,59].

We have compared the differential gene expression profiles of these genes in the GU^high^ versus GU^low^ subset. The results are summarized in Table 2. The deregulation pattern, as delineated above, supports the notion that the GU^high^ subset exhibits the characteristics of senescent cells.

As a reference, the expression profiles of these genes in the CD34+ cells derived from older (>60 years) versus young adults (<33 years) were compared and depicted in Table 2. While the correlation to the expression profiles of genes involved in cell cycle arrest was highly significant for GU^high^ versus GU^low^, the corresponding comparison between old versus young adult CD34+ cells showed only a trend in some and statistical significance in a few of the interrogated genes (Table 2).

In murine models of aging, the INK4/ARF locus seems to play an important role within the context of sustained p53 activation [19]. Both p16^INK4a^ and ARF are encoded by the CDKN2A gene, and p15^INK4b^ encoded by the CDKN2B gene. ARF inhibits the ubiquitin ligase MDM2, subsequently contributing to increased levels of P53 [60]. We could not detect any significant difference in comparing the expression levels of CDKN2A or CDKN2B between GU^high^ versus GU^low^ subsets, nor between old versus young CD34+ cells. This is in agreement with reports from the literature that senescence in murine cells is more dependent on p19^Arf^ than that in human cells [61,62].

### 7.2. Senescence-Associated Secretory Phenotype (SASP)

The production of a combination of inflammatory factors called senescence-associated secretory phenotype (SASP) is another characteristic of senescent cells [29]. Whereas the combination of secreted factors depends on the specific cell type and the inducer, some of the key effectors of the SASP seem to be shared by senescence cells of diverse origins. NOTCH1, CCAAT/enhancer-binding protein beta (CEBPB) and nuclear factor kappa B (NFKB) are some of the major shared regulators [63,64]. Cytokines such as IL1A, IL1B, and IL6 are also involved. The differential expressions of some of these genes in GU^high^ versus GU^low^ subsets were interrogated and the results summarized in Table 2.

The differential gene expression analysis revealed that the GU^high^ showed a consistently higher expression of genes involved in SASP: NOTCH1, CEBPA, CEBPB, NFKB1, NFKB2, IL1B, and IL1R2. The results are compatible with the notion that the GU^high^ subset comprises cells of the SASP phenotype. As a reference, the expression profiles of these genes in the CD34+ cells derived from older (>60 years) in comparison to young adults (<35 years) are also listed in Table 2.

### 7.3. TP53, Apoptosis, and Pro-Survival Pathways

Another major feature of senescent cells is their survival despite activated DNA damage responses, increased levels of SASP inflammatory cytokines, and cell cycle arrest—all factors that might have induced apoptosis. Senescent cells are, however, equipped with anti-apoptotic and pro-survival mechanisms [65,66,67]. Using transcript array analysis, Zhu et al. demonstrated upregulation of negative regulators of apoptosis and pro-survival gene sets in senescence versus non-senescent cells [68]. Using a pre-adipocyte radiation-induced senescence model, increased expression of pro-survival networks around EFNB-1, EFNB-3, PI3KCD, p21 (CDKN1A), PAI-1 (SERPINE1), PAI-2 (SERPINB2), BCL-xL, and MCL-1 has been shown [68]. Among these genes, the roles of upregulations of BCL2, BCL-xL (BCL2L1), and MCL1 as prominent anti-apoptotic factors in murine HSPC have been extensively examined [25,62,68].

Our scRNA-seq studies showed that the GU^high^ subset of CD34+ cells expressed significant upregulation of these anti-apoptotic and pro-survival genes (see Table 2) [54].

## 8. Gene Set Enrichment Analysis (GSEA)

We applied gene set enrichment analysis (GSEA; version 4.2.1, Broad Institute, Inc., Massachusetts Institute of Technology, and Regents of the University of California) to identify pathways and processes that were coordinately up- or downregulated with senescence [69]. The “Hallmarks” gene sets and “ENSEMBL Gene identifiers MSigDB.v7.4chip” were applied to determine the differential expression profiles in biological processes associated with GU^high^ versus GU^inter^ and/or GU^low^ subpopulations [70].

In the GU^high^ phenotype, 40 of 50 gene sets interrogated by “Hallmarks” are upregulated, 30 of which significantly enriched at a nominal *p* value of <1%, and 35 gene sets are significantly enriched at *p* = <5%. In Table 3, we have summarized the four most prominently elevated pathways associated with senescence [54].

Gene sets that were significantly upregulated include (1) G2M checkpoint; (2) MTORC1 signaling; (3) inflammatory response; (4) apoptosis (Table 3). Gene sets significantly downregulated include interferon gamma and interferon alpha response [54].

The upregulation of the pathway G2M checkpoint in the GU^high^ subset is compatible with the cycle arrest that is characteristic for senescent cells. Inflammatory response is closely linked to the phenotype SASP. Mammalian target of rapamycin (MTOR) activity has been shown to be increased in HSPC from old mice compared to those from young mice and suppression of MTOR by rapamycin prolonged lifespan in mice [71,72]. Finally, upregulation of apoptosis/pro-survival as well as P53 pathways in aged murine HSPC has been reported to be characteristic [41]. The simultaneous upregulation of these Hallmark gene sets in the GU^high^ subset has therefore provided further evidence for the senescence phenotype in this population.

We have performed the same GSEA algorithm comparing the CD34+ cells derived from old versus young adults. While upregulation of all the gene sets involved in the aforementioned biological processes was highly significant in comparing the GU^high^ versus GU^low^ subset, only the upregulation of the “P53 Pathway” in the aged CD34+ cells reached statistical significance at *p* = <5% [49,54]. The proportion of senescent cells in the whole CD34+ population is probably too small for an appropriate comparison. This observation again provides evidence that the GU^high^ subset in the HSPC compartment is highly enriched for senescent cells.

## 9. GSEA Using “Aging Signature Gene Sets” as Reference

Svendsen et al. surveyed 16 published and unpublished transcriptome datasets from comparisons of aged and young murine HSPC [20]. They were able to identify a list of 220 genes that were consistently deregulated in aging murine HSPC. Their meta-analysis and re-analysis of the published data suggested that this gene set collection of aging signature (AS) remained stable and robust. They concluded that this AS can be used as a reference for differentially expressed genes upon HSPC aging.

Using this gene set as a reference, we have performed additional GSEA to compare the differential expression profiles of GU^high^ versus GU^low^, as well as GU^high^ versus GU^inter+low^. The results are summarized in Table 3. The GU^high^ subset showed high scores, whereas the GU^low^ subset low scores for the gene set “aging signature” [20]. This is also true when the GU^low^ subpopulation was compared with GU^inter^ (data not shown).

This additional comparison using the “aging signature” has again confirmed that the GU^high^ subpopulation is highly enriched for senescence in the human HSPC compartment. Another remarkable finding is also that the expression profile of the GU^low^ subset has maintained that of primitive and young adult HSPC and did not change with age in all these studies. This is compatible with the reports from Svendsen et al. that a subset of murine HSPC have maintained a “young-like” phenotype.

## 10. The Causative Role of Senescent Cells in Aging and Their Elimination

Senescent cells have been shown to play a causative role in age-related pathologies in mice [71,72,73,74]. The abundance of senescence cells among the total number of somatic stem cell compartments may be relatively low, i.e., up to a maximum of 15% in some tissues. Their impact is, however, probably magnified by the secretion of pro-inflammatory cytokines that constitute the SASP [64,73,74]. The SASP in turn accounts for many of the consequences of senescent cell accumulation, leading to dysfunction of the stem cell pool as well as to diseases of the respective tissue. Elimination of or a reduction in senescent cells from the stem cell pool may eventually ameliorate functional decline of HSPC.

In animal models, genetic ablation of senescent cells (p16^Ink4a^-expressing cells) by activating a drug-inducible pro-apoptotic gene has been shown to attenuate the onset of age-related pathologies of progeroid mice [24]. In a subsequent study by the same group, p16^Ink4a^-expressing cells were consistently eliminated by the use of a transgene INK-ATTAC that induced apoptosis in the respective cells upon administration of a specific drug [25]. The authors concluded that therapeutic removal was able to extend healthy lifespan in mice. Since then, a number of studies have provided further evidence that reducing the burden of senescent cells by as little as 30% could ameliorate age-related disabilities and chronic diseases [26,27,75].

Senescent cells in mice have been shown to be well equipped with anti-apoptotic and pro-survival mechanisms [24,62,68,76]. Anti-apoptotic proteins such as BCL2, BCL-XL, and BCL-W are upregulated and seem to be essential for the survival of senescent HSPC in animal models [24,25,76]. This has inspired the development of inhibitors of anti-apoptotic factors as “senolytic” agents to eliminate senescent cells [25,68,76,77]. Inhibitors of anti-apoptotic factors, e.g., ABT263 and ABT737, are able to selectively deplete senescent HSPC and clearance of the latter has led to rejuvenation not only of HSPC, but also to improved cardiovascular function, reduced osteoporosis and extended health span in naturally aging mice [76,78].

Our finding of significantly elevated glucose metabolism in the senescent population of human HSPC compartment indicates that this subset is dependent on elevated central carbon metabolism for survival, in analogy to the increased expression of anti-apoptotic factors. Modulation of the glycolytic pathways may therefore represent another therapeutic principle for senolysis treatment.

## 11. Conclusions

Comprehensive and unbiased proteomics studies of the HSPC have provided evidence that aging of the HSPC compartment is characterized by elevated glycolysis. Whereas other significant changes found with this proteomics approach, such as an increase in differentiation bias towards the myeloid lineage, in DNA repair, in cellular metabolism, and a decrease in proteins responsible for lymphoid development and for stabilization of DNA replication have been described in murine transcriptomics studies, the enhanced central carbon metabolism activity in aged human HSPC is novel [23].

Subsequent studies have demonstrated that the increase in glycolytic enzyme level is caused by the expansion of a HSPC subset that has become more glycolytic than the others and not on a per cell basis. Provided with this knowledge, we developed a method to isolate the HSPC according to their glucose metabolic levels in three distinct categories: the GU^high^, GU^inter^ and GU^low^ subsets. The GU^high^ subset is coupled with differentiation bias towards myeloid lineages. GSEA of the transcriptomes of the GU^high^ versus GU^low^ subset, or GU^high^ versus GU^inter^ subsets, have demonstrated that the gene sets for cell cycle arrest, MTORC1 signaling, inflammatory response, and anti-apoptosis pathways are significantly upregulated in the GU^high^ population. Applying the transcriptomic “aging signature” gene set proposed by [20], the GU^high^ subset achieves high scores for this “aging signature”.

Most of the current knowledge about the properties of senescent cells in the HSPC compartment is based on experiments in cultured cells and in murine models of aging. There are no specific markers or marker constellations that could enable us to identify and enrich senescent cells for mechanistic studies. With the separation of HSPC according to glucose metabolic levels, we have shown that we are able to enrich the senescent population in the GU^high^ subset in the human HSPC compartment of the bone marrow. As the viability and functional integrity of this subset are well preserved, these cells may serve as starting material for further mechanistic characterization of senescent HSPC.

In analogy to the Warburg effect in cancer cells, our data have provided strong evidence for the dependency of senescent HSPC on elevated central carbon metabolism as well as on MTORC1 pathway for survival. During development and aging of HSPC, drastic metabolic shifts to meet the demand of hematopoiesis during transition occurs [79]. The glycolytic and MTORC1 pathways integrate inputs from nutrient and growth signals to regulate general cellular processes such as protein and lipid synthesis, autophagy, and metabolism [80]. In this respect, MTOR has been shown to regulate the senescence-associated secretory phenotype (SASP) and senescent growth arrest [80,81]. Based on upstream signaling of MTORC1, a relationship between carbohydrate consumption and MTORC1 activity has been demonstrated, specifically through the insulin growth factor pathway [82]. Multiple studies have demonstrated that caloric restriction can retard the aging decline [83]. Hence, caloric restriction, or agents that modulate the glycolytic pathway such as Metformin may modulate glycolysis and MTORC1 pathways and eliminate the senescent population within the HSPC compartment.

Senescent cells and cancer cells share many common properties. Agents that block the apoptotic pathways that cancer cells are dependent on have been shown to be effective as senolytic drugs in aged mice. ABT-263 and ABT-737 are examples that target the B cell lymphoma 2 (BCL-2) protein family members. These drugs are, however, associated with toxic side effects such as neutropenia and thrombocytopenia. Targeting the central carbon metabolism or the closely related MTORC1 signaling pathway may offer better alternatives for developing senolysis strategies.

In summary, our series of studies are unique in the following aspects. First of all, almost all present-day knowledge on aging of HSPC and most proof-of-principle investigations for elimination of senescent cells have been gained from studies in mice. Without any doubt, great advances have been achieved, but the knowledge must be validated in the human system. Our comprehensive transcriptome and proteome datasets have contributed to bridge this gap. As delineated in this review, many of the principal mechanisms of senescence in mice can be confirmed in human system and yet there are differences. Another significant aspect is the discovery of the close association of elevated central carbon metabolism with senescence. Thus far, isolation and collection of senescent cells have been extremely difficult as specific markers or marker constellations for their identification have yet to be developed. This is specially the case in human HSPC. The enrichment of HSPC, with all the characteristics of senescence by glucose metabolism, in conjunction with single-cell high-throughput technology, may represent an important stepping stone towards accurate visualization, collection and tracking of senescence in human bone marrow.

## Figures and Tables

**Figure 1 ijms-23-03028-f001:**
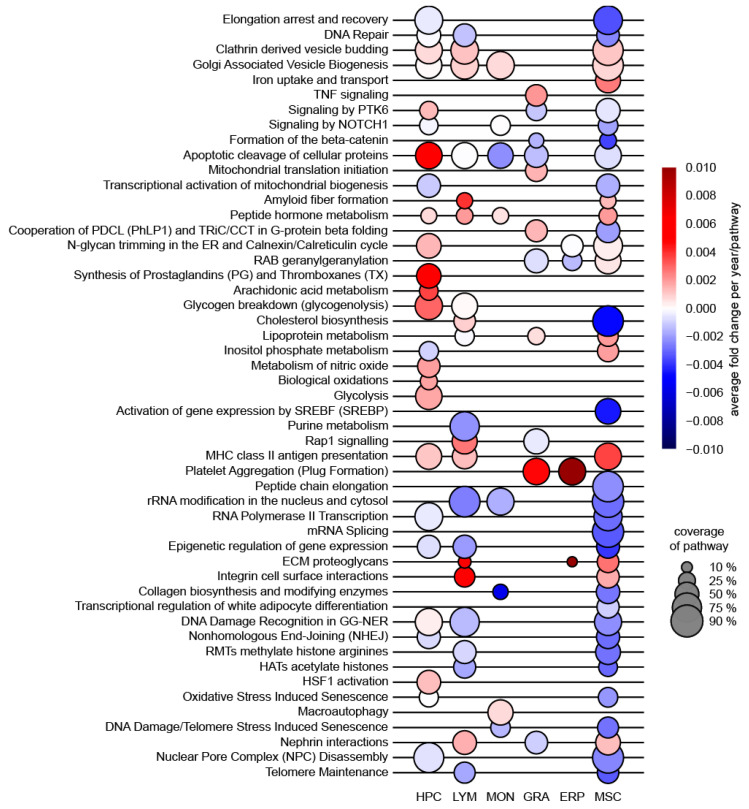
Prominent proteome changes upon aging in six individual cell types in human bone marrow (modified from Hennrich et al. [23]). Selected pathways that were significantly altered, i.e., either up- or downregulated with age are depicted. The size of each circle represents the percentage coverage of a pathway with proteins assessed by LF quantification in our dataset. The color indicates the direction of change, with red being upregulated and blue being downregulated with age.

**Figure 2 ijms-23-03028-f002:**
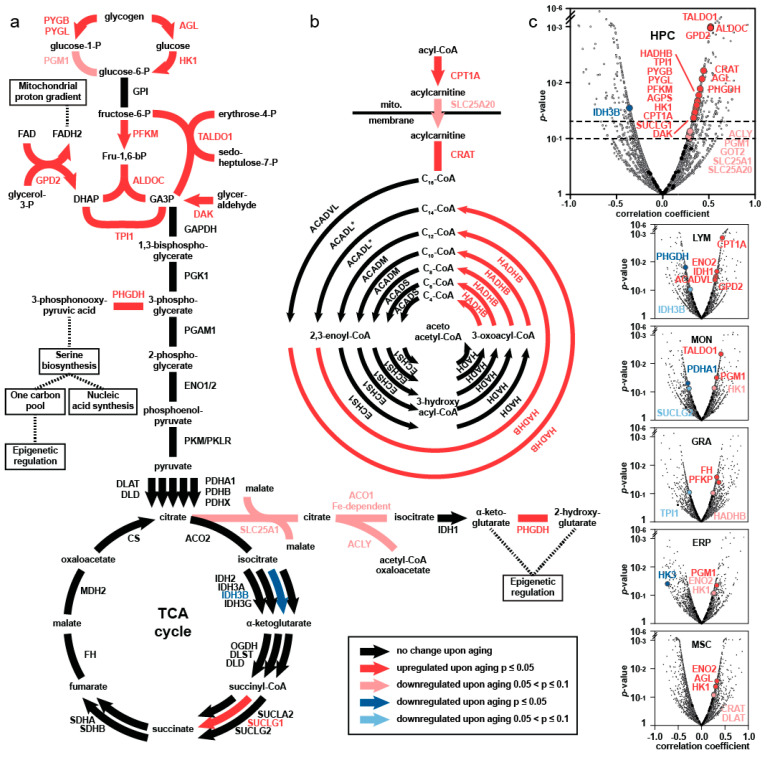
Changes in central carbon metabolism of HSPC upon aging (modified from Hennrich et al. [23]). (**a**) Glucose metabolism pathway and the tricarboxylic acid (TCA) cycle are depicted. The respective enzymes are shown in capital letters and the color encodes for changes upon aging. Glycogen is broken down and the products are fed into glycolysis. Glucose-6 phosphate is metabolized to dihydroxyacetone-phosphate (DHAP) and glyceraldehyde-3-phosphate (GA3P). The enzymes GPD2 and DAK that feed the pool of DHAP and GA3P increase upon aging as well as TALDO1. (**b**) A specific set of enzymes involved in the mitochondrial beta-oxidation of fatty acids increase in abundance with age. (**c**) Volcano plots of all proteins quantified in the respective cell populations. The elevation of enzymes involved in the central carbon metabolism is found only in the HSPC.

**Figure 3 ijms-23-03028-f003:**
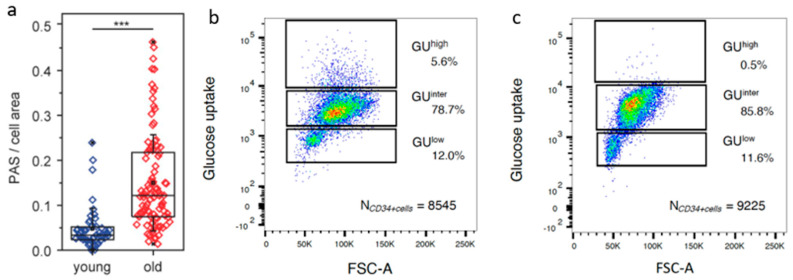
Glycogen accumulation and glucose uptake capacity of the CD34+ HSPC (modified from Poisa et al. [49]). (**a**) Results of the semi-quantitative assessment of intensity of PAS reaction as surrogate marker for glycogen in CD34+ cells from young versus old human subjects. The average glycogen content was significantly higher in old subjects than young subjects < 35 years (one-sided *t*-test, *p* = 1.2 × 10^−6^, *** *p* < 0.001). (**b**,**c**) Glucose uptake capacity of the total CD34+ cells was assessed by Cayman’s Glucose Uptake Assay Kit. Incubation for 30 min with 1.75 μg/mL 2-NBDG yielded a dose-dependent uptake of glucose into the CD34+ cells. The latter could then be separated according to their respective levels of glucose uptake by a FAC-Sorter into three distinct subpopulations according to the glucose uptake (GU) capabilities: GU^low^, GU^inter^, and GU^high^. (**b**) An example of separation of CD34+ cells from a human subject > 50 years according to GU capacity. (**c**) An example of separation of CD34+ cells from a young subject (<35 years). In young subjects only, GU^inter^ and GU^low^, but scarcely any GU^high^ cells could be detected.

**Table 1 ijms-23-03028-t001:** Fraction of GU^high^, GU^inter^, and GU^low^ cells from young (≤35 years) versus older (>50 years) healthy human subjects. ns: not significant. (Slightly modified from Poisa-Beiro et al. [49].

CD34+ Subsets	Young (*n* = 3) in %	Old (*n* = 7) in %	*p*
GU^high^	1.7 ± 1.5	5.4 ± 3.5	0.02
GU^inter^	66.5 ± 36.9	66.4 ± 22.5	ns
GU^low^	31.8 ± 36.7	28.2 ± 21.7	ns

**Table 2 ijms-23-03028-t002:** Differential expressions of genes involved in biological processes. (a) Cell cycle arrest; (b) senescence-associated secretory phenotype; (c) apoptosis and pro-survival.

Gene Symbols	GU^high^ vs. GU^low^ *p* Values	Old vs. Young *p* Values
**(a) Cell cycle arrest**		
TERF2	***	0.073
POT1	0.082	*
CHEK1	***	0.633
CHEK2	***	0.717
TP53	*	*
CDKN1A (P21 CIP1)	***	0.498
CDKN2A (P15 INK4A)	0.244	0.103
RB1	***	0.193
**(b) Senescence associated secretory phenotype**		
NOTCH1	***	0.530
CEBPA	***	**
CEBPB	***	0.281
NFKB1	*	0.451
NFKB2	*	0.330
IL1B	***	*
IL1R1	0.081	0.101
IL1R2	**	na
**(c) Apoptosis and pro-survival**		
BCL2	***	0.0899
BCL2L1	***	0.681
BCL3	***	0.18
MCL1	***	*

Genes that are downregulated are depicted in blue, and those that are upregulated in red. Significance codes: *** *p* < 0.001, ** *p* < 0.01, and * *p* <0.05.

**Table 3 ijms-23-03028-t003:** Gene set enrichment analysis (GSEA) of the GU^high^ versus GU^low^ subsets.

	*GS TITLE (HALLMARK ANALYSIS)*	*SIZE*	*ES*	*NES*	*NOM* *p-val*	*FDR* *q-val*	*RANK AT MAX*	*LEADING* *EDGE*
a	** *G2M CHECKPOINT* **	198	0.59	2.39	0.000	0.000	2893	tags = 49%, list = 13%, signal = 56%
b	** *MTORC1 SIGNALING* **	198	0.43	1.92	0.000	0.003	4682	tags = 45%, list = 22%, signal = 57%
c	** *INFLAMMA-TORY RESPONSE* **	172	0.40	1.75	0.008	0.010	5872	tags = 47%, list = 27%, signal = 64%
d	** *APOPTOSIS* **	152	0.34	1.55	0.014	0.030	4055	tags = 30%, ist = 19%, signal = 37%
e	***SVENDSEN* et al. *(2021)***	176	0.46	2.03	0.000	0.000	4453	tags = 49%, list = 13%, signal = 56%

In the GU^high^ phenotype, of all 50 gene sets interrogated by “Hallmarks”, 30 gene sets are significantly enriched at a nominal *p* value of <1%, and 35 gene sets are significantly enriched at *p* = <5%. In Table 3, we have summarized the four most prominently elevated pathways associated with senescence (a) G2M checkpoint; (b) mTORC1 signaling; (c) inflammatory response; (d) apoptosis; (e) using gene set “aging signature” proposed by Svendsen et al. 2021. ES = enrichment score; NES = normalized enrichment score; NOM *p*-val = nominal *p* value; FDR = false discovery rate.

## Data Availability

The mass spectrometry proteomics data have been deposited at the ProteomeXchange Consortium via the PRIDE partner repository with the dataset identifier PXD007048. Raw data for both the single-cell RNA-seq and bulk RNA-seq experiments have been deposited in the Gene Expression Omnibus (GEO), database under accession code GSE115353, as described in detail in References [23,49].

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
