# Peer review of "Glucose Metabolism and Aging of Hematopoietic Stem and Progenitor Cells"

_ijms, 2022, doi:10.3390/ijms23063028_

Round 1

Reviewer 1 Report

In this manuscript, the authors show comprehensive proteomics studies of human hematopoietic stem and progenitor cells have revealed that aging of the HSPC compartment is characterized by elevated glycolysis. In addition to deregulations found in murine transcriptomics studies, such as an increased differentiation bias towards the myeloid lineage, alterations in DNA repair, and a decrease in lymphoid development.

This manuscript contains interesting remarks, is well written and organized and with a very updated bibliography. The quality of the tables is very good and exhaustive.

There is just one point that the authors to be considered before the publication.

the review focuses a lot of attention by repeating data from previous manuscripts and leaves out a lot of aspects that are only hinted at. for example, the authors do not explain the role of the niche in inducing the senescence of HSPC (DOI: 10.1111/acel.12933. 10.1038/s41598-018-26693-x.).

Therefore it would be advisable for the authors to reinforce with a more updated bibliography their manuscript. 

Author Response

Reviewer 1

In this manuscript, the authors show comprehensive proteomics studies of human hematopoietic stem and progenitor cells have revealed that aging of the HSPC compartment is characterized by elevated glycolysis. In addition to deregulations found in murine transcriptomics studies, such as an increased differentiation bias towards the myeloid lineage, alterations in DNA repair, and a decrease in lymphoid development.

We deeply appreciate the critical comments of Reviewer 1. We would also like to emphasize that in addition to the proteomics studies of aging of HSPC, we have put the major emphasis on the results of single cell transcriptomics studies in this review, proving that elevated glucose metabolism is a significant feature of aging HSPC, i.e. senescent cells. This is found only in a small subset of HSPC, i.e. maximally 7 to 8% in human subjects >70 years. The biological significance of this finding is twofold: (1) This property might be exploited to enrich, isolate, and track senescent cells in human bone marrow; and (2) Targeting glucose metabolism may represent another avenue for “senolytic” therapy strategies.

This manuscript contains interesting remarks, is well written and organized and with a very updated bibliography. The quality of the tables is very good and exhaustive. 

We would like to thank the reviewer for this very constructive and positive comment.

There is just one point that the authors to be considered before the publication:
the review focuses a lot of attention by repeating data from previous manuscripts and leaves out a lot of aspects that are only hinted at. for example, the authors do not explain the role of the niche in inducing the senescence of HSPC (DOI: 10.1111/acel.12933. 10.1038/s41598-018-26693-x.).

We would like to thank the reviewer for drawing our attention to aspects that we have left out, such as the role of the niche in inducing senescence. We agree that changes in the niche, and the interactions of the HSPC with the aging niche definitely play a role in senescence of the HSPC. Indeed, changes in the niche are so complex that these should be dealt with in a separate review and far beyond the scope of the present manuscript. The focus of our article is on metabolic regulations and aging of HSPC. Nevertheless, we have made several changes in the revised manuscript to cover this aspect, and have added three up-dated references as suggested by the Reviewer.
a. The importance of the niche is mentioned on P.2, lines 49-54, and lines 70-75.
b. References have been up-dated (11, 12, 13 in the revised version), as suggested by Reviewer 1.

Reviewer 2 Report

The present manuscript aims at dissecting the metabolic alterations associated with hematopoietic stem and progenitor cells (HSPC) upon aging. The manuscript is of good quality and relevant for designing future modulation strategies for addressing senescent HSPC. However, it is not clear whether it is a review or simple discussion of previous findings from the group, coupled to further gene expression analysis.

Indeed, as a review it covers a very narrow topic of discussion and does not explore how these findings can be relevant for therapeutic strategies. As a research article (e.g. short communication) it lacks a clear explanation of the methodology employed in this study.

Therefore, it would be essential to re-write the manuscript based on one of these article types.

Author Response

Reviewer 2

The present manuscript aims at dissecting the metabolic alterations associated with hematopoietic stem and progenitor cells (HSPC) upon aging. The manuscript is of good quality and relevant for designing future modulation strategies for addressing senescent HSPC. However, it is not clear whether it is a review or simple discussion of previous findings from the group, coupled to further gene expression analysis.

We thank Reviewer 2 for raising this issue and for the constructive suggestions. We agree that our findings are relevant for designing modulation strategies for eliminating senescent HSPC. We have now rewritten diverse parts of the manuscript to emphasize that this is a review article with special focus on evidence gained from single cell transcriptomics studies. For example: on page 2, lines 70-75;  on page 11, lines 415-419.

Indeed, as a review it covers a very narrow topic of discussion and does not explore how these findings can be relevant for therapeutic strategies….Therefore, it would be essential to re-write the manuscript based on one of these article types.

As recommended by Reviewer 2, we have now re-written several paragraphs, e.g. in the Section “The causative role of senescent cells in aging and their elimination”, (P. 11, lines 415-419), and another paragraph in the Section “Conclusions” (P.12, lines 449-462) to highlight the relevance of our findings for identifying the molecular mechanisms of metabolic changes upon aging of HSPC, and their potential importance for future therapeutic interventions.

Round 2

Reviewer 1 Report

Thank you for your comments.

Author Response

We would like to thank Reviewer 1 for the very constructive and positive comment. 

Reviewer 2 Report

The authors have discussed some aspects of the relevance of dissecting the metabolic alterations associated with hematopoietic stem and progenitor cells (HSPC) upon aging. Nevertheless, no other major revisions were made, being most figures modified versions of previously published data. It Is therefore still required that the authors perform major changes of the manuscript structure:

1) consider citing previous findings from the group without providing experimental details (e.g. p6-7, lines 195 - 261) and results (figure 1,2,3 and table 1);

2) focus on the two "original" tables In the manuscript (table 2 and 3) and describe the methodology employed In their preparation In table caption.

Author Response

In accordance with the recommendations by Reviewer 2, we have

  1. ..shortened the paragraphs between line 176 (page 5) to line 235 (previously line 261). Table 1 has been deleted but Figures 1, 2 and 3 are essential for capturing the major developments and arguments for the main focus of this review. The changes are highlighted in yellow.
  2. ..focused on the two "original" Tables (now Table 1 and Table 2), and have described the methodology employed in their preparation in the Caption of Table 1 (lines 270 to 281, page 8). 

We have therefore performed all the major changes in the structure of the manuscript as suggested by Reviewer 2. 

Round 3

Reviewer 2 Report

The authors have addressed the major questions raised in previous review report.